# Transcriptome analysis of sevoflurane exposure effects at the different brain regions

Hiroto Yamamoto[1,2ᵒ], Yutaro Uchida[1ᵒ], Tomoki Chiba[1ᵒ], Ryota Kurimoto[1], Takahide Matsushima[1], Maiko Inotsume[1], Chihiro Ishikawa[3], Haiyan Li[3], Takashi Shiga[3,4], Masafumi Muratani[5], Tokujiro Uchida[2]*, Hiroshi Asahara[1,6]*

1 Department of Systems BioMedicine, Tokyo Medical and Dental University (TMDU), Tokyo, Japan, 2 Department of Anesthesiology, Tokyo Medical and Dental University (TMDU), Tokyo, Japan, 3 Graduate School of Comprehensive Human Sciences, University of Tsukuba, Tsukuba, Japan, 4 Department of Neurobiology, Faculty of Medicine, University of Tsukuba, Tsukuba, Japan, 5 Department of Genome Biology, Faculty of Medicine, University of Tsukuba, Tsukuba, Japan, 6 Department of Molecular Medicine, The Scripps Research Institute, La Jolla, California, United States of America

ᵒ These authors contributed equally to this work.
* asahara@scripps.edu, asahara.syst@tmd.ac.jp (HA); uchida.mane@tmd.ac.jp (TU)

## Abstract

### Backgrounds

Sevoflurane is a most frequently used volatile anesthetics, but its molecular mechanisms of action remain unclear. We hypothesized that specific genes play regulatory roles in brain exposed to sevoflurane. Thus, we aimed to evaluate the effects of sevoflurane inhalation and identify potential regulatory genes by RNA-seq analysis.

### Methods

Eight-week old mice were exposed to sevoflurane. RNA from medial prefrontal cortex, striatum, hypothalamus, and hippocampus were analysed using RNA-seq. Differently expressed genes were extracted and their gene ontology terms were analysed using Metascape. These our anesthetized mouse data and the transcriptome array data of the cerebral cortex of sleeping mice were compared. Finally, the activities of transcription factors were evaluated using a weighted parametric gene set analysis (wPGSA). JASPAR was used to confirm the existence of binding motifs in the upstream sequences of the differently expressed genes.

### Results

The gene ontology term enrichment analysis result suggests that sevoflurane inhalation upregulated angiogenesis and downregulated neural differentiation in each region of brain. The comparison with the brains of sleeping mice showed that the gene expression changes were specific to anesthetized mice. Focusing on individual genes, sevoflurane induced *Klf4* upregulation in all sampled parts of brain. wPGSA supported the function of KLF4 as a transcription factor, and KLF4-binding motifs were present in many regulatory regions of the differentially expressed genes.

**Data Availability Statement:** The data are available from DDBJ with DRA accession number DRA010292.

**Funding:** This work was supported by Japan Society for the Promotion of Science KAKENHI (URL:https://www.jsps.go.jp/english/index.html, Grant Nos. 20H00547, 19KK0227, 18K19603 and 15H02560 to HA) and AMED-CREST from AMED (URL: https://www.amed.go.jp/en/index.html, Grant No. JP20gm0810008 to HA). The funders had no role in study design, data collection and analysis, decision to publish, or preparation of the manuscript.

**Competing interests:** The authors have declared that no competing interests exist.

## Conclusions

*Klf4* was upregulated by sevoflurane inhalation in the mouse brain. The roles of KLF4 might be key to elucidating the mechanisms of sevoflurane induced functional modification in the brain.

## Introduction

Sevoflurane is the most frequently used volatile anesthetic in general anesthesia. Some reports discussed the perioperative adverse effects of sevoflurane, such as emergence agitation, postoperative delirium, and cognitive disorders, although whether anesthetics themselves cause perioperative adverse effects is still controversial [1–3]. Several membrane receptors such as the γ-aminobutyric acid type A receptor, nicotinic AchR, hyperpolarization-activated cyclic nucleotide-gated channels have been reported to be potential targets of sevoflurane [4–9]. However, receptor-based molecular mechanisms have not sufficiently explained these phenomena. Furthermore, although some reports have evaluated the effects of sevoflurane using transcriptome analysis, these studies focused only on limited parts of the brain. Hayase et al. reported that the increase in dopamine activity in the hippocampus due to inhalation of sevoflurane might be related to postoperative nausea, and Mori et al. reported circadian gene variations in the suprachiasmatic nucleus after sevoflurane inhalation [10, 11]. However, we thought that by comparing many regions at once and extracting genes that might play common role in all regions, we could focus on genes that are important with regard to the whole brain.

In this study, medial prefrontal cortex (MPFC), hippocampus, striatum, and hypothalamus were chosen as targets of the analysis, as these parts were frequently used for evaluating the effects of volatile anesthetics [12–15]. Differently expressed genes (DEGs) and enriched gene groups were compared between the four parts of the brain and we applied the same analysis to the transcriptome array data of sleeping mice to identify specific gene expression changes in brains exposed to sevoflurane. Finally, we evaluated the effects of the transcription factors on their target genes using wPGSA and confirmed the existence of consensus-binding motifs in the upstream sequences of DEGs. Herein, we report sevoflurane-induced gene expression change patterns in the mouse brain and that KLF4 emerged as a specific transcription factor that potentially promoted angiogenesis and induced the appearance of undifferentiated neural cells.

## Materials and methods

### Approval for the animal experiments

All the animal experiments in this study were conducted in accordance with the Guidelines for Proper Conduct of Animal Experiments (Science Council of Japan) and approved by the Center for Experimental Animals of Tokyo Medical and Dental University. (Approval No.A2017-131A)

### Experimental conditions and preparation of brain samples

Eight-week old mice (C57BL/6J) were purchased from Sankyo Labo (Tokyo, Japan) and Oriental Yeast (Tokyo, Japan). Six mice were assigned into two groups, the control (n = 3) and sevoflurane inhalation groups (n = 3). For the sevoflurane group, the mice were put in a box with 2.5% sevoflurane / 40% oxygen for 3 hours. The body temperature was measured and

sustained within the range of ±0.5˚C by using a body warming machine. For the control group, the mice were put in a box with normal air and stayed in the box without food or water for 3 hours. After the treatments, all the mice were immediately killed through cervical dislocation, and their whole brains were removed. The brain samples were cut into 2 mm slices, and the medial prefrontal cortex, striatum, hippocampus, and hypothalamus were punched out, referring to the methods of Ishikawa et al [16].

### RNA extraction from brain tissue sections and RNAseq analysis

RNA was extracted from brain tissue sections by using TRIZOL (ThermoFisher, Waltham, MA, USA) and 500ng total RNA was used for the subsequent preparation. RNA-seq libraries were prepared with a rRNA-depletion kit (E6310, New England Biolabs Japan, Tokyo, Japan) and a directional library synthesis kit (E6310, New England Biolabs Japan). The RNA libraries were sequenced using NextSeq500 High-output kit v2 for $2 \times 36$ base reads.

### Mapping FASTQ data and calculating gene expressions

The adapters in the FASTQ files were trimmed using the TrimGalore software (https://www.bioinformatics.babraham.ac.uk/projects/trim_galore/). The FASTQ files were mapped to mouse genomes (mm10) by using the STAR software [17] (https://github.com/alexdobin/STAR), and the amount of each transcript was calculated with the RSEM software [18] (https://github.com/deweylab/RSEM).

### Extracting DEGs on iDEP.91

The counted data were transformed with EdgeR ($\log_2$ [counts per million (CPM) + 4]), and principal components analysis (PCA) plots were depicted. DEGs were extracted using DESeq2. All these steps were performed with iDEP.91 [19] (http://bioinformatics.sdstate.edu/idep/). Venn-diagrams were used to depict the upregulated and downregulated DEGs.

### Sequencing data

The raw sequencing data were submitted to the DNA Data Bank Japan (DDBJ: http://www.ddbj.nig.ac.jp) under accession No. DRA010292.

### Gene ontology term enrich analysis using Metascape

The extracted DEGs were analysed with Metascape [20] (http://metascape.org/gp/index.html#/main/step1). A gene ontology (GO) term enrichment analysis was performed, and a Circos plot was drawn.

### Extraction of DEGs from sleeping mice

The transcriptome array data of the unbound fractions of immunoprecipitation for the cerebral cortices of waking or sleeping mice (GSE69079) were used in the analysis [21]. The expression data were normalized, and DEGs were selected using DESeq2 in iDEP. 91. Venn-diagrams were used to depict the DEGs of the MPFCs of anesthetized mice and cerebral cortices of the sleeping mice.

## Weighted Parametric Gene Set Analysis (wPGSA) of DEGs in the brains of mice that inhaled sevoflurane

For the fold changes data, the expression changes of the target genes of each transcription factor were calculated and its activity (T-score) was estimated using a weighted parametric gene set analysis (wPGSA [22]: http://wpgsa.org/). Expressed transcription factors in the brain were extracted from the analysed data. The transcription factors with T-scores of > 2.0 and false discovery rates (FDRs) of < 0.1 were regarded as active transcription factors, while those with T-scores of < - 2.0 and FDR of < 0.1 were regarded as suppressive transcription factors. Moreover, the transcription factors included in the DEGs were extracted from active and suppressive transcription factors and, regarded as responsible for anesthetic effects.

## Histological and immunohistochemical analysis

Mouse brain was fixed in 4% paraformaldehyde overnight at 4°C, and embedded in paraffin. Sections of 4μm in thickness were stained. Immunohistochemical staining was performed using a Vectastain ABC-AP Rabbit IgG Kit (AK-5001, VECTOR LABORATORIES, INC., CA, USA) and Vector Red (SK-5100, VECTOR LABORATORIES, INC.) according to the manufacturer's instructions. Anti-KLF4 antibody (1/100 dilution) (NBP2-24749, Novus Biologicals, CO, USA) was used as the primary antibodies.

## Western blotting analysis for brains exposed to sevoflurane

Proteins were collected from hippocampus with lysis buffer (10 mM Tris-HCl, 2%SDS) with protease inhibitor (WAKO, Osaka, Japan). 50μg (for KLF4) or 20μg (for ACTB) of proteins were separated by SDS-PAGE followed by semi-dry transfer to a PVDF membrane. Membranes were blocked for 1h with Blocking-One (Nacalai Tesque, Kyoto, Japan), reacted with primary antibody for KLF4 (4038S, CST, MA, USA) or ACTB (010–27841, WAKO) at 4°C overnight, rinsed and reacted with ECL mouse IgG HRP-conjugated whole antibody (GE Healthcare, IL, USA) or rabbit IgG HRP-conjugated whole antibody (GE Healthcare). The blot was developed using the ECL Select Western Blotting Detection Reagent (GE Healthcare).

## Detection of the consensus-binding motifs of Klf4 in the upstream sequences of DEGs

The consensus-binding motifs of KLF4 were referred from JASPAR (http://jaspar.genereg.net/). The 1000-bp upstream sequences of the DEGs annotated with the GO terms "angiogenesis" and "head development" were analysed using JASPAR and the existence of KLF4 binding motifs was confirmed. We regarded the motifs with scores of > 8 as candidate binding motifs for KLF4.

## Statistical analyses

In extracting DEGs from RNA-seq data and differently activating transcription factors from the wPGSA analysed data, we considered FDRs of < 0.1 as statistically significant.

# Results

## Genome-wide transcriptome analysis for the brains of mice that inhaled sevoflurane

To investigate the sevoflurane-induced gene expression changes in the brain, three 8 week old male mice that inhaled sevoflurane for 3 hours or the control mice were killed, and their brains

were removed. The brain tissue samples were cut into 2 mm slices, and four parts of the brain (hippocampus, hypothalamus, medial prefrontal cortex and striatum) were punched out for RNA extraction. We performed a genome-wide transcriptional analysis with next generation sequencing, and confirmed the proper RNA extraction from each brain area by the PCA plot (Fig 1A, S1A and S1B Fig). DEGs were extracted on the basis of the criteria of FDR < 0.1 (S1 Table). Among the upregulated DEGs, 100, 109, 33, and 314 were expressed in the striatum, MPFC, hypothalamus and hippocampus, respectively. Among the downregulated DEGs, 93, 121, 18, and 502 were expressed in the striatum, MPFC, hypothalamus, hippocampus, respectively (Fig 1B–1E, S2 Table). The highest number of DEGs was found in the hippocampus; and the lowest number in the hypothalamus. These results suggest that the gene expression in the hippocampus was the most-influenced and that in the hypothalamus was the least-influenced by sevoflurane inhalation.

(B)~(E) FASTQ files were mapped using STAR, differently expressed genes (DEGs) were extracted using iDEP.91 and MA-plots were drawn for the striatum (B), medial prefrontal cortex (C), hypothalamus (D), and hippocampus (E).

To compare the upregulated DEGs in the different parts of the brain, a Venn-diagram was drawn (Fig 2A). Thirteen common upregulated genes found in all sampled parts of the brain are shown in Fig 2B. Sevoflurane inhalation upregulated transcription factors such as *Klf4* in all sampled parts (Fig 2B). The expression level of *Klf4* was >2.5 times higher than that in the control mice. Furthermore, to investigate the differences of upregulated DEGs between the different parts of the brain, a GO term enrichment analysis was performed using Metascape [20]. The Circos plot drawn using Metascape showed similarities in the upregulation patterns of the gene expressions in the four parts of the brain (Fig 2C). As shown in the heatmap, sevoflurane inhalation caused the upregulation of genes annotated as "angiogenesis" and "response to wounding" in all parts (Fig 2D). The transcription factors KLF4 and KLF2, as well as EDN1, CCN1, and ADAMTS1, were annotated to the GO terms "angiogenesis" and "response to wounding" (S3 Table).

Next, downregulated DEGs were compared between the four parts of the brain, and a Venn-diagram was drawn (Fig 3A). The common downregulated DEGs among each part of brain was only the Banp gene (Fig 3B). Furthermore, to compare the downregulated DEGs between the four parts of the brain, a GO term enrichment analysis was performed with Metascape. As shown in the Circos plot, enriched GO terms were similar among the different parts of the brain (Fig 3C). Moreover, the heatmap showed that sevoflurane inhalation downregulated the genes annotated as "head development" in all sampled parts of brain, and those annotated as "axon development" or "synapse organization" in several parts (Fig 3D and S4 Table).

For identifying specific gene expression changes induced by sevoflurane inhalation, a comparison was made with the transcriptome array data of the cerebral cortices of sleeping mice as the resembling state [21]. We chose the data of MPFCs exposed to sevoflurane as an equivalent part to the cerebral cortices of the sleeping mice. We extracted DEGs using the same method in our experiments. Regarding the comparison between the gene expression changes in the cerebral cortices of the sleeping mice and those of the waking mice, the sleeping mice had 477 upregulated DEGs and 3572 downregulated DEGs (S5 and S6 Tables). As shown in the Venn diagrams, there were 5 common upregulated DEGs and 45 common downregulated DEGs were found between the sevoflurane-anesthetized and sleeping mice (S2A and S2B Fig, S7 Table). Moreover, by comparing genes upregulated and downregulated in all parts of the brain exposed to sevoflurane, we found that all the genes except *Edn1* were completely expressed differently (S2C and S2D Fig).

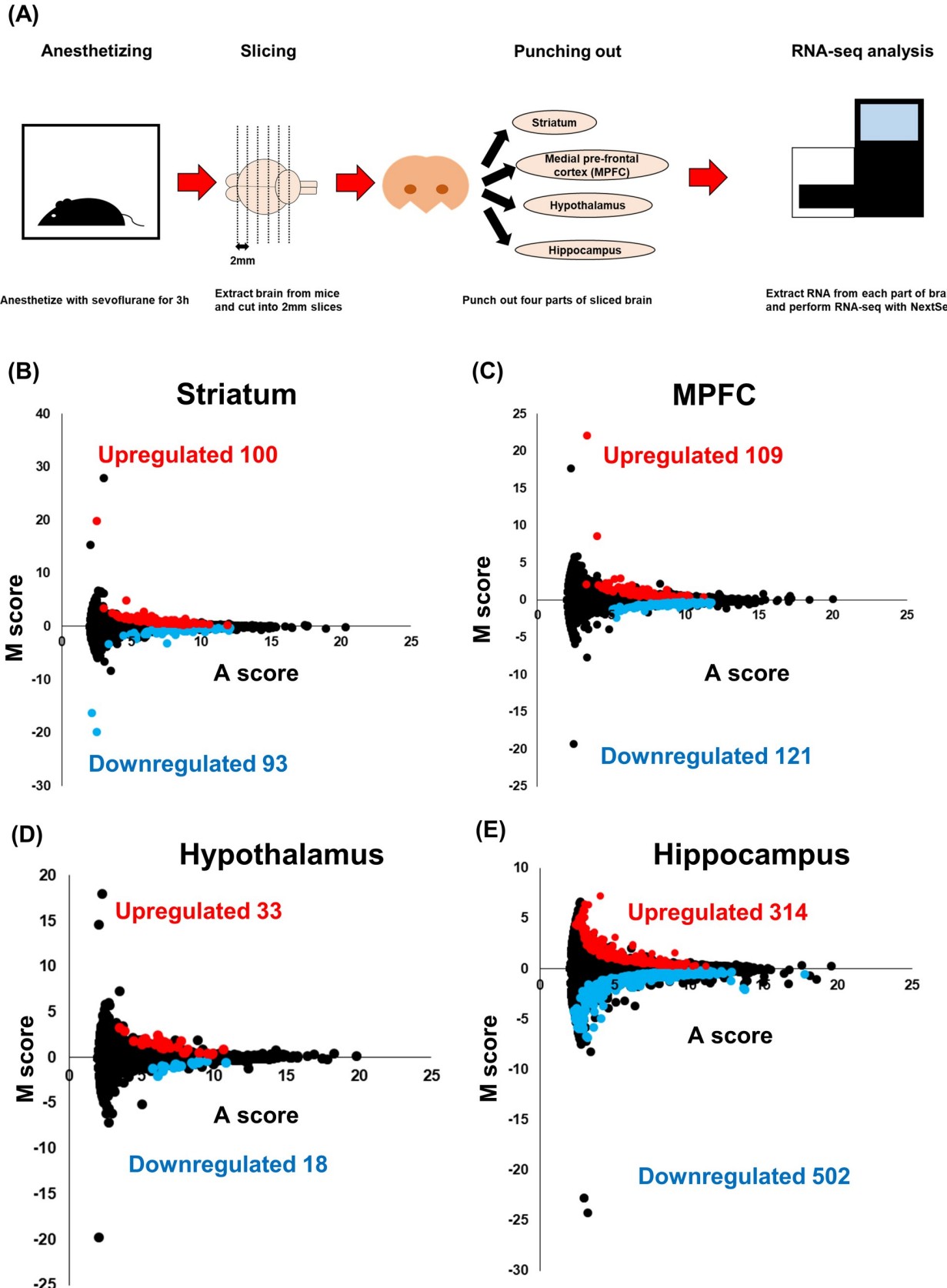

**Fig 1. RNA-seq analysis for brains exposed to sevoflurane.** (A)The workflow of the RNA-seq analysis for the anesthetized mice. After anesthetizing with 2.5% of sevoflurane and 40% oxygen for 3 hours, the brains were removed and sliced into 2 mm pieces. The striatum, medial prefrontal cortex (MPFC), hypothalamus and hippocampus were punched out. RNA was extracted from the punched out samples and RNA-seq was performed using NextSeq500.

## Identification of KLF4 as a candidate of key transcriptional regulator in brain exposed to sevoflurane

As represented by KLF4, sevoflurane induced changes in the expressions of many transcription factors from the analysis of DEGs. Therefore, we hypothesized that sevoflurane changed the activities of specific transcription factors in each part of the brain. To verify this hypothesis, we utilized the wPGSA method [22], with which evaluated the expression changes of the target genes for each transcription factor by using T-scores. A positive T-score means that the transcription factor functions as an activator, while a negative T score means that it functions as a repressor. We regarded transcription factors with both |T-score| > 2.0 and FDR < 0.1 as functional transcription factors. With the wPGSA method, 34, 3, 3, and 1 transcription factors in the MPFC, striatum, hypothalamus, and hippocampus were estimated as activators, respectively. Ninety-three, 188, 113, and 168 transcription factors in the MPFC, striatum, hypothalamus, and hippocampus were estimated as repressors, respectively (Fig 4A, 4C, 4E and 4G; S8 Table). Moreover, we identified activators and repressors included in the DEGs, inferring that they particularly functioned owing to the induction by sevoflurane. In the MPFC, the target genes of *Klf4*, *Klf2*, and *Per2* were upregulated, while those of *Atf4* and *Taf1* were downregulated (Fig 4B). Likewise, in the striatum, the target genes of 5 transcription factors were downregulated, and in the hypothalamus, the target genes of KLF4 were downregulated (Fig 4D and 4F). Finally, in the hippocampus, the target genes of 14 transcription factors were downregulated (Fig 4H). These results indicate that KLF4 plays some important roles in gene expression in brains exposed to sevoflurane. To validate the upregulation of KLF4, we performed immunohistochemical analysis for the cerebral cortex and hippocampus. As a result, we observed that the expression of KLF4 was strongly upregulated in the nucleus of cells in the cerebral cortex of mice exposed to sevoflurane. On the other hand, nucleus in neural cells of hippocampus in both control mice and mice exposed to sevoflurane showed high expression of KLF4, and no significant changes were observed in immunohistochemical analysis (S3A Fig). Based on these results, we performed western blotting analysis to validate the upregulation of KLF4 in the hippocampus, showing a certain upregulation of KLF4 (S3B Fig).

Even *Klf4* was upregulated in all four parts of the brain, it worked as an activator in the MPFC, and as a repressor in the other three parts of brain. KLF4 was reported to function as both as an activator and a repressor, and this result might reflect the different transcriptional roles of KLF4 between each part of brain [23]. Moreover, the expression of the same *Klf* family gene, *Klf2*, was also upregulated in the MPFC and functioned as an activator, while *Klf5* and its target genes were downregulated in the striatum and hippocampus. These results indicate the possibility of cooperative functions between the same *Klf* family genes.

Finally, we confirmed the existence of consensus sequences of KLF4 in the DEGs of important functions. The consensus-binding motif sequence of murine KLF4 was GGG(T/C)G(G/T)GGC according to JASPAR (http://jaspar.genereg.net/). On JASPAR, we searched the candidate binding sites of KLF4 in 1000bp upstream sequences for upregulated DEGs annotated GO of "angiogenesis", and downregulated DEGs annotated GO of "head development". As shown in the pie charts, 82.7% of the GOs of the upregulated DEGs annotated as "angiogenesis", and 82.5% of the GOs of the downregulated DEGs annotated as "head development" had consensus-binding motifs in their 1000-bp upstream sequences (Fig 5A and 5B, S9 Table).

**(A)**

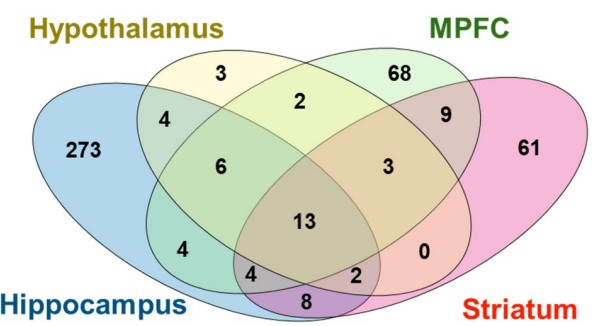

**(B)**

| Common Up-regulated Genes (Expression fold change) | | | | |
|---|---|---|---|---|
| | **Hippocampus** | **MPFC** | **Hypothalamus** | **Striatum** |
| Edn1 | 3.110 | 2.243 | 2.096 | 1.941 |
| Apold1 | 2.379 | 1.833 | 2.206 | 2.755 |
| Maff | 2.285 | 1.763 | 1.652 | 1.867 |
| Ccn1 | 2.262 | 2.955 | 2.420 | 1.781 |
| Klf4 | 1.599 | 2.007 | 1.718 | 2.043 |
| Akap12 | 1.583 | 1.402 | 0.875 | 1.459 |
| Nes | 1.476 | 1.506 | 1.763 | 1.253 |
| Gm49839 | 1.350 | 1.325 | 1.613 | 1.705 |
| Tinagl1 | 1.282 | 1.465 | 1.472 | 1.196 |
| Arrdc2 | 0.969 | 1.199 | 1.130 | 1.236 |
| Adamts1 | 0.674 | 1.238 | 1.483 | 1.603 |
| Txnip | 0.614 | 0.946 | 0.904 | 0.618 |
| Clic4 | 0.501 | 0.859 | 0.440 | 0.469 |

**(C)**

**(D)**

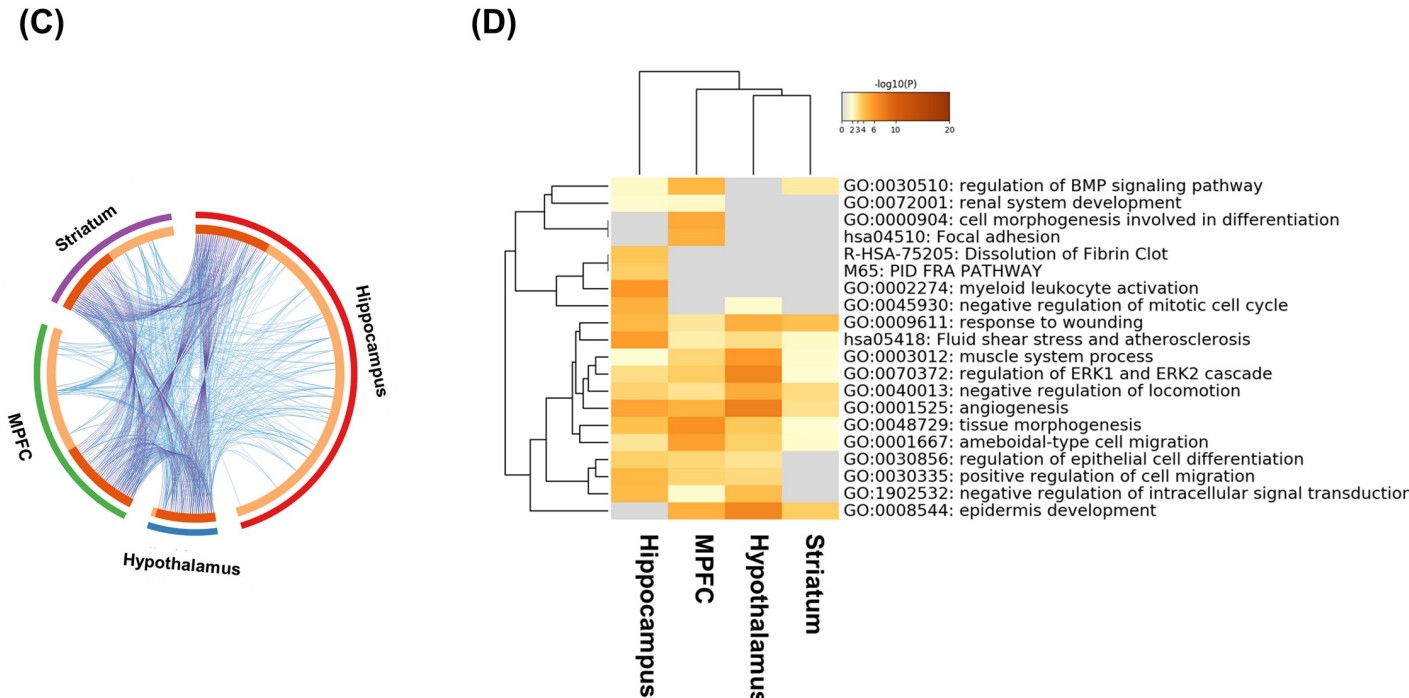

**Fig 2. Analysis of the upregulated Differently Expressed Genes (DEGs) in each part of the brain.** (A) Venn-diagram for the upregulated differently expressed genes (DEGs) in each part of brain. (B) The 13 genes commonly upregulated in the four parts of the brain and the fold changes (log2) for each gene. (C) Circos plot for the Metascape analysis of upregulated DEGs. The purple line links the same gene that is shared by multiple gene lists. The blue lines link the different genes where they fall into the same ontology term. (D) Heatmap for the gene ontology term analysis of the upregulated DEGs.

These results indicate that KLF4 has the potential to regulate the transcription of genes related to angiogenesis and neural development, which might contribute to vascular neogenesis and the appearance of undifferentiated neural cells (Fig 5C).

## Discussion

In this study, our group performed a genome-wide transcriptional analysis for the brains of mice that inhaled sevoflurane. Results of our analyses suggest that sevoflurane induced both

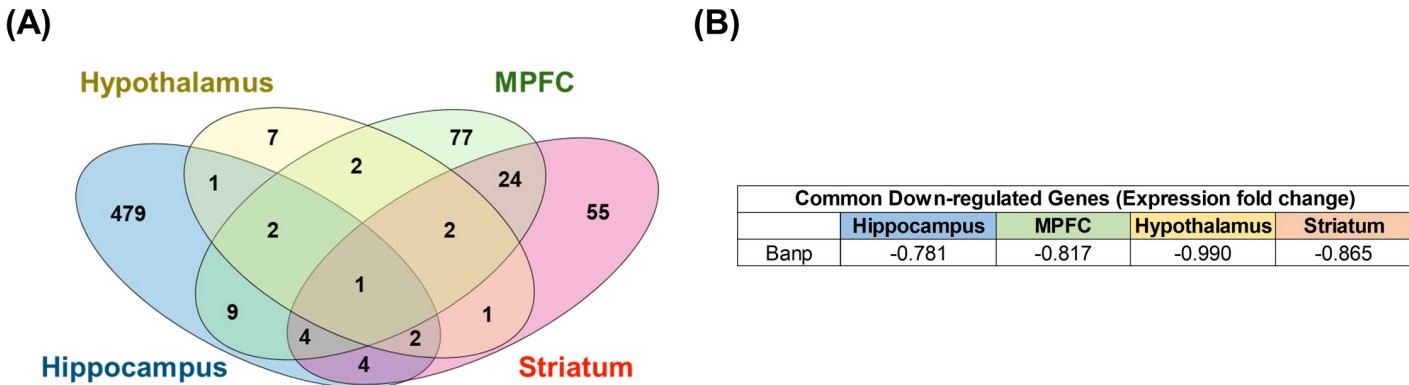

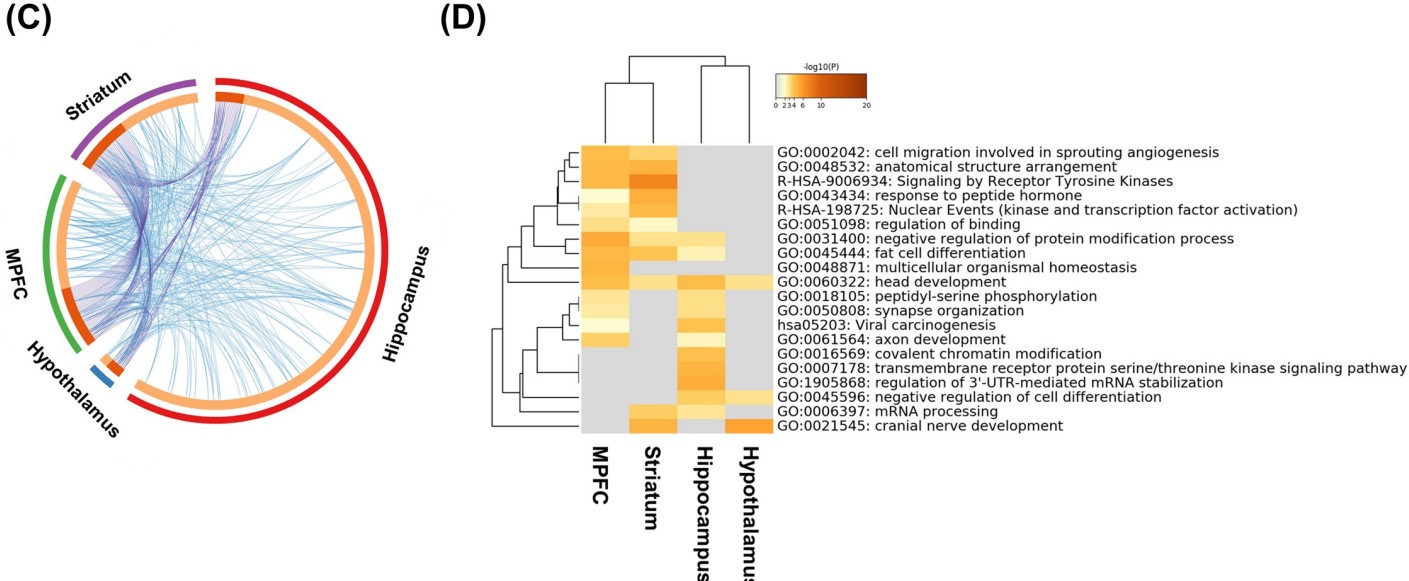

**Fig 3. Analysis of the downregulated differently expressed genes for each part of the brain.** (A) Venn-diagram for the downregulated differently expressed genes (DEGs) in each part of brain. (B) Commonly downregulated gene in the four parts of the brain and its fold change (log2). (C) Circos plot for the Metascape analysis of the upregulated DEGs. The purple line links the same gene that are shared by multiple gene lists. The blue lines link the different genes where they fall into the same ontology term. (D) Heatmap for gene ontology terms analysis of the upregulated DEGs.

angiogenesis and the appearance of undifferentiated neural cells in all sampled parts of brain. These changes in gene expression were not observed in the brains of sleeping mice, and seemed specific to brains exposed to sevoflurane. The transcription factor *Klf4* was commonly upregulated in all sampled brain, and the results of the wPGSA and motif analysis suggest that KLF4 is a key transcriptional regulator of the angiogenesis and appearance of undifferentiated neural cells.

KLF4 is known as an essential regulator of the initialization of iPS cells, or so-called "Yamanaka factor" [24]. Moreover, the redundant and cooperative functions between KLF2 and KLF5 were reported to be important for sustaining the undifferentiated state of ES cells [25]. Thus, KLF2, KLF4, and KLF5 are known to be fundamental factors for sustaining undifferentiated states. Considering the upregulation of Nestin, which is a specific marker of

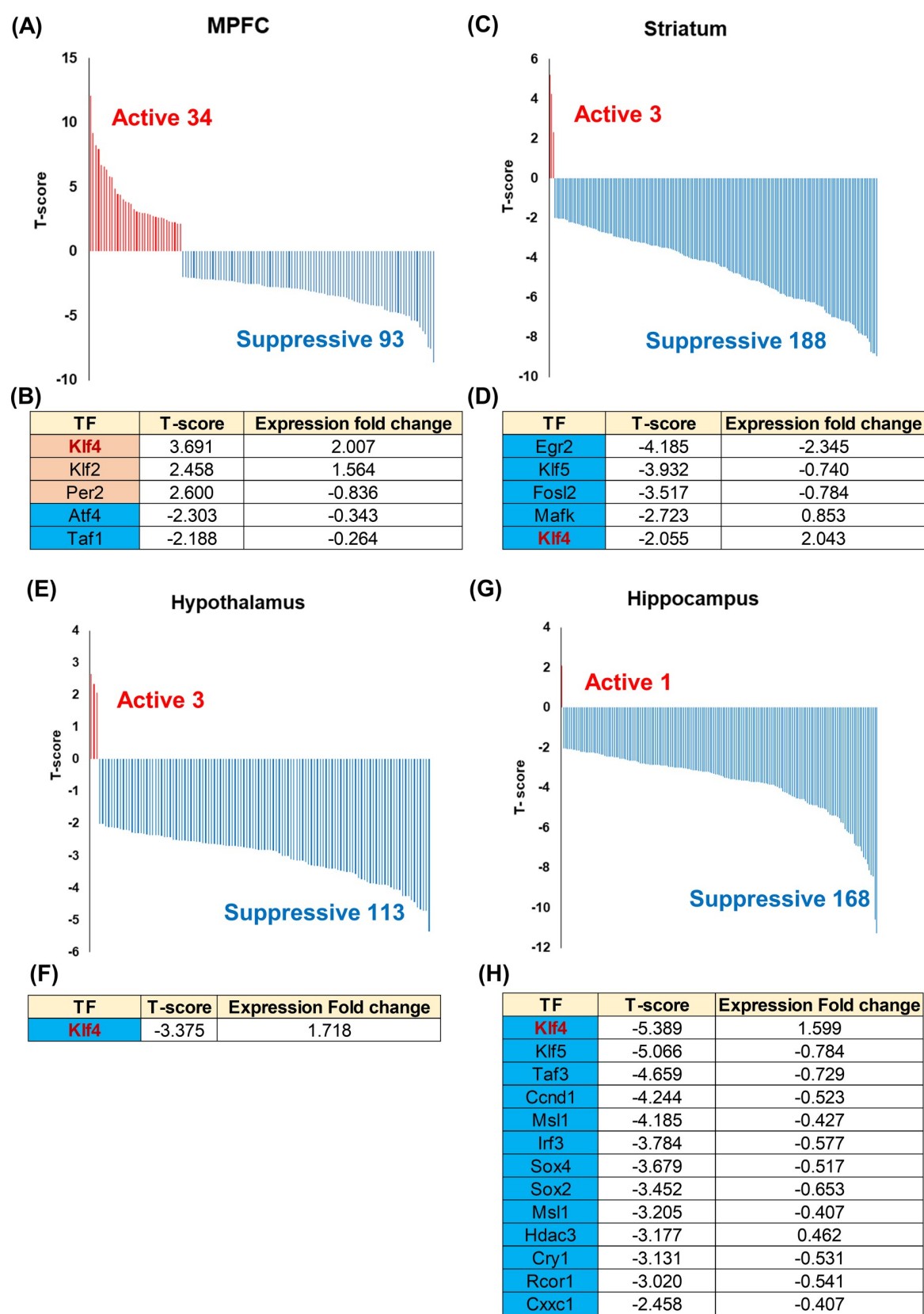

**Fig 4. Estimation and comparison of the relative activities of the transcriptional factors.** (A)-(H) Weighted parametric gene set analysis (wPGSA) of the fold changes in each part of the brain. Transcription factors (TFs) with T-scores of > 2.0 or < -2.0 were identified and the distributions of the T-scores of the medial prefrontal cortex (MPFC) (A), striatum (C), hypothalamus (E), and hippocampus (G) were drawn. Furthermore, the transcription factors included in differently expressed genes (DEGs) were identified and the tables of the T-scores and expression fold changes for MPFC (B), striatum (D), hypothalamus (F), and hippocampus (H) were made.

undifferentiated neural cells, and the decreasing expression of genes associated with neural differentiation, sevoflurane inhalation seemed to cause the appearance of undifferentiated neural cells by the *Klf* family genes.

In the previous report, sevoflurane administration decreased the cerebral blood flow in a statistical parametric mapping analysis [26]. Other reports also indicated that sevoflurane inhalation caused permeability of the brain-blood barrier induced the plasma influx into the brain parenchyma, possibly causing postoperative delirium and cognitive decline [27]. Our results that show the upregulation of genes encoding angiogenesis and the appearance of undifferentiated cells were potentially related with these functional changes in the brain caused by sevoflurane. In this context, KLF4 seemed to be the key regulator of these genes, and precise analyses of the roles of KLF4 might be key to unveiling the mechanism of the sevoflurane anesthesia-induced postoperative functional modification of the brain.

Detailed analysis between anesthesia and sleep is difficult because of the different experimental conditions, but at least in this comparison, gene expression changes in the brain exposed to sevoflurane showed a pattern that was very different from that of sleep. Especially KLF4 seemed to function specifically by sevoflurane inhalation. The roles of KLF4 seemed to differ among the parts of the brain in our wPGSA. KLF4 has multiple functions, including as activators and repressors, and work context- dependently [23, 28, 29]. Furthermore, our analysis revealed that KLF4 had potentials to upregulate genes related to angiogenesis and downregulate neural differentiation. The variable activity of KLF4 might reflect the differences of these activities between the parts of the brain. For a precise understanding of the specific roles of KLF4 induced by sevoflurane, chromatin immunoprecipitation analysis of KLF4 and histone markers, such as H3K9me3 and H3K27Ac in each part of brain are needed. Furthermore, experimental methods that combine single-cell RNA-seq and location information such as Slide-seq may provide more useful information [30]. Nevertheless, our analysis results indicated the importance of KLF4 as a candidate regulator of the effects caused by sevoflurane inhalation.

Our report, which focuses on the changes of transcription factors, provides original and novel approaches for analysing the effects of anesthetics in brain. This is the first report to evaluate the effects of sevoflurane inhalation, focusing on the activities of transcription factors. As a limitation of this study, only three of samples were used. However, we concluded that increasing replicates did not significantly change the results because of the high reproducibility between triplicates, supported by the PCA plot (S1B Fig). Furthermore, we could not exclude the possibility of the effect of the hypoxic condition caused by the respiratory depression induced by sevoflurane [31]. However, our experimental condition (2.5% sevoflurane in 40% oxygen for 3 hours) is common setting in experiments for studies on the effects of sevoflurane on the brain. None of the genes related to hypoxic reaction, including *Hif1a* and *Arnt*, were detected in our analyses of gene expression changes, supporting the exclusion of the possibility of hypoxia in our experimental conditions (S1 Table). Conversely, oxygen saturation might have been higher in the anesthetized group than in the control group, which was allowed to spend time in room air, and since we did not measure oxygen saturation, it is possible that subtle differences in oxygen saturation existed and that this might have affected the results. Oxygen saturation assessment in mice may provide more reliable results. Nevertheless, our

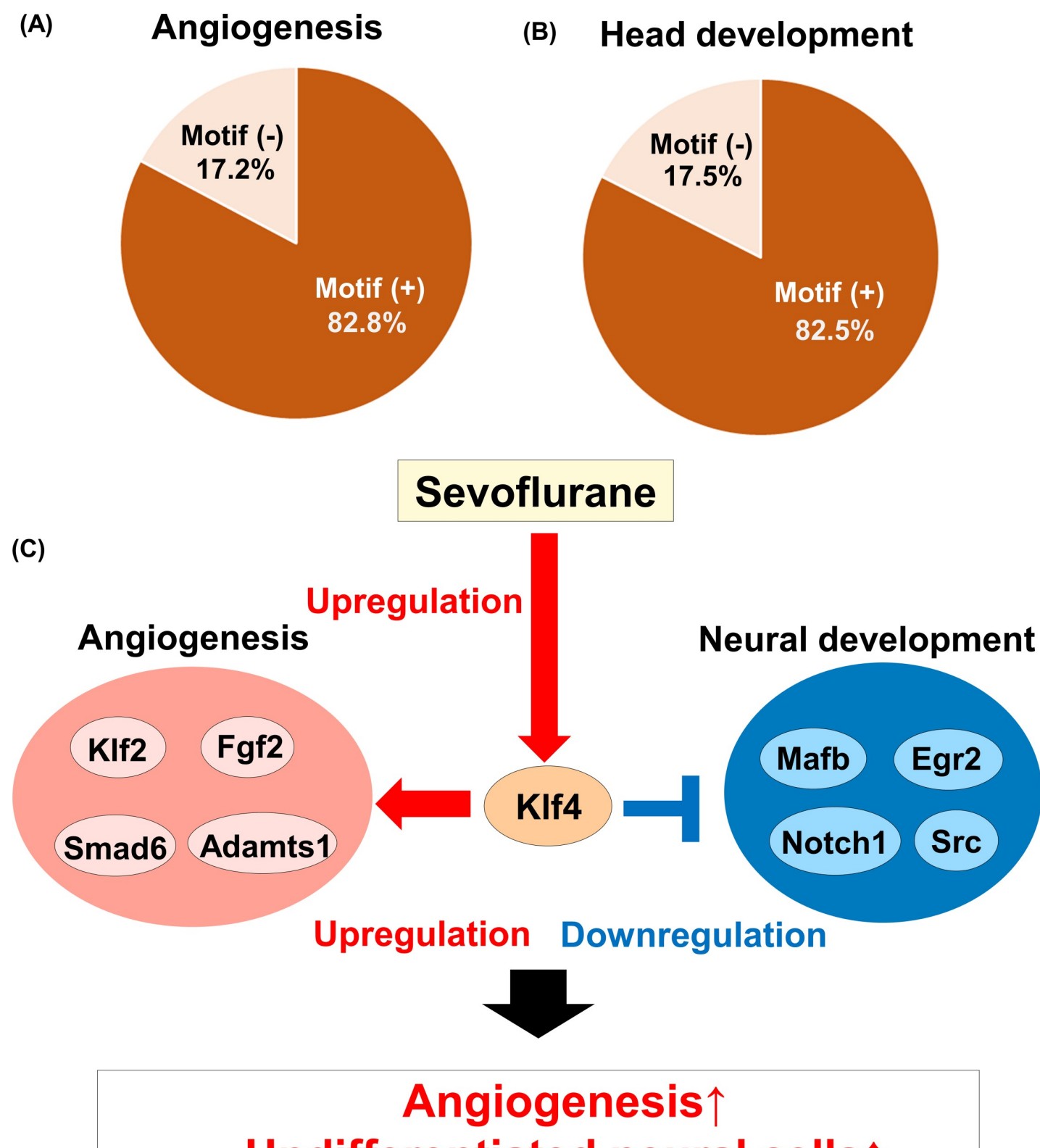

**Fig 5. Comparison of the activities of the transcription factors between the brains of the anesthetized and sleeping mice.** (A) Pie chart of the existence of KLF4-binding motifs in the 1000-bp upstream sequences of the genes annotated to the GO term "angiogenesis". (B) Pie chart of the existence of KLF4-binding motifs in the 1000-bp upstream sequences of the genes annotated to the GO term "head development". (C) Estimated mechanism of the effects of sevoflurane on the brain.

strategy should include better choices for obtaining the whole image of brain activities under anesthetized condition.

In conclusion, the results of our genome-wide transcriptional analysis of the brains of mice that inhaled sevoflurane suggest the upregulation of angiogenesis and appearance of undifferentiated neural cells. Moreover, we identified KLF4 as a potential regulator of the effects induced by sevoflurane inhalation.

## Supporting information

**S1 Fig. RNA-seq analysis for anesthetized brain.** (A)The distribution of $\log_2$ ((count per million) +4) after normalization. (B)PCA-plot for RNA-seq data.
(TIF)

**S2 Fig. Comparison of Differently Expressed Genes (DEGs) between the brains of the anesthetized and sleeping mice.** (A, B) DEGs were extracted from the transcriptome array data of the cortical cortices of the sleeping mice. The DEGs in the medial prefrontal cortex of the mice that inhaled sevoflurane and those in the cortical cortices of the sleeping mice were compared. The Venn-diagrams for the upregulated (A) and downregulated DEGs (B) are shown. (C) Table of the expression fold change (log2) of the genes commonly upregulated in the four parts of the brain of the mice that inhaled sevoflurane. (D) Table of expression fold changes (log2) of the genes commonly downregulated in the four parts of brain of the mice that inhaled sevoflurane.
(TIF)

**S3 Fig. Immunohistochemistry and western blotting for hippocampus of brains exposed to sevoflurane.** Representative image of immunohistochemical analysis of KLF4 for cerebral cortex and hippocampus of mice exposed to sevoflurane. Western blotting for hippocampus of brains exposed to sevoflurane.
(TIF)

**S1 Table. Gene expression data for all the genes in all parts of the brain of mice that inhaled sevoflurane.** $\log_2$ (read counts per million +4) of all the genes of all parts of the brain from the RNAseq analysis data by iDEP91 are shown.
(XLSX)

**S2 Table. Gene lists of differently expressed genes in each part of the brain.** The gene names and expression fold change data (sevoflurane group vs control group) of the hippocampus, hypothalamus, striatum, and medial prefrontal cortex are shown.
(XLSX)

**S3 Table. Lists of genes and gene ontology terms of upregulated differently expressed genes.** Metascape analysis was performed for upregulated differently expressed genes. The gene ontology (GO) terms, their *p* values and genes annotated to each GO terms are shown in the table.
(XLSX)

**S4 Table. Genes and gene ontology term lists of downregulated differently expressed genes.** A Metascape analysis was performed for downregulated differently expressed genes. The gene ontology (GO) terms, their *p* values and genes annotated to each GO terms are shown.
(XLSX)

**S5 Table. Expression and fold change data for each gene from the transcriptome array data of the cortical cortices of sleeping mice.** The gene names, transcriptome array data and expression fold change data (sleeping group vs control group) from GSE69079 are shown.
(XLSX)

**S6 Table. Lists of the differently expressed genes in the cerebral cortices of sleeping mice.** The gene names and each expression fold change data (sleeping group vs control group) for the upregulated and downregulated genes are shown.
(XLSX)

**S7 Table. Comparison of the gene expression fold changes of the common differently expressed genes between mice that inhaled sevoflurane and sleeping mice.** The gene names and each expression fold change data for the common upregulated and downregulated genes (sevoflurane group vs control group and sleeping group vs control group) are shown.
(XLSX)

**S8 Table. Lists of the transcription factors and their T-scores from the wPGSA for each part of brain.** The activities of the transcription factors (TFs) in the medial prefrontal cortex, striatum, hypothalamus, and hippocampus were calculated using the wPGSA analysis. The T-scores of the transcription factors are shown.
(XLSX)

**S9 Table. List of the predicted binding motifs of Klf4 in the upstream sequences of the differently expressed genes.** The predicted binding motifs of KLF4 for the 1000-bp upstream sequences of the differently expressed genes were identified using JASPAR.
(XLSX)

## Acknowledgments

We specially thanked to technical supports for Kana Shishido and Tomomi Kato, and are grateful to all staffs of the Department of Systems BioMedicine at Tokyo Medical and Dental University (TMDU) for their support and discussion.

## Author Contributions

**Conceptualization:** Hiroto Yamamoto, Yutaro Uchida, Tomoki Chiba, Takahide Matsushima, Tokujiro Uchida, Hiroshi Asahara.

**Data curation:** Hiroto Yamamoto.

**Formal analysis:** Yutaro Uchida, Tomoki Chiba, Masafumi Muratani.

**Funding acquisition:** Hiroshi Asahara.

**Investigation:** Hiroto Yamamoto, Yutaro Uchida, Tomoki Chiba, Maiko Inotsume, Chihiro Ishikawa, Haiyan Li, Takashi Shiga, Masafumi Muratani, Tokujiro Uchida.

**Methodology:** Hiroto Yamamoto, Yutaro Uchida, Tomoki Chiba, Takahide Matsushima, Maiko Inotsume, Chihiro Ishikawa, Haiyan Li, Takashi Shiga, Masafumi Muratani, Tokujiro Uchida.

**Project administration:** Hiroshi Asahara.

**Supervision:** Tomoki Chiba, Ryota Kurimoto, Takahide Matsushima, Tokujiro Uchida, Hiroshi Asahara.

**Writing – original draft:** Hiroto Yamamoto, Yutaro Uchida, Tokujiro Uchida.

**Writing – review & editing:** Hiroto Yamamoto, Yutaro Uchida, Tomoki Chiba, Ryota Kurimoto, Takahide Matsushima, Takashi Shiga, Masafumi Muratani, Tokujiro Uchida, Hiroshi Asahara.

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
