## [Decision Letter · Decision Letter 0]

7 Sep 2020

PONE-D-20-21488

Transcriptome analysis of sevoflurane exposure effects at the different brain regions

PLOS ONE

Dear Dr. Asahara,

Thank you for submitting your manuscript to PLOS ONE. After careful consideration, we feel that it has merit but does not fully meet PLOS ONE’s publication criteria as it currently stands. Therefore, we invite you to submit a revised version of the manuscript that addresses the points raised during the review process. 

Your manuscript has been carefully evaluated by four external reviewers with expertise in this field. They are basically positive, but raised several concerns below. Especially, they suggest that changes in protein expression of KLF4 by sevoflurane exposure, and roles of KLF4 up-regulation on the angiogenesis or differentiation of neural cells, should be demonstrated. Further consideration of appropriate controls, description and Interpretation of the results, and data availability is also required. These critical suggestions should be addressed for further consideration. Because conclusions are not presented in an appropriate fashion and are not supported by the data, this manuscript cannot be recommended for publication in PLoS ONE in its current form. 

We look forward to receiving your revised manuscript.

Kind regards,

Wataru Nishimura, M.D., Ph.D.

Academic Editor

PLOS ONE

Journal Requirements:

Reviewers' comments:

Reviewer's Responses to Questions

**Comments to the Author**

1. Is the manuscript technically sound, and do the data support the conclusions?

Reviewer #1: No

Reviewer #2: Yes

Reviewer #3: Partly

Reviewer #4: Yes

2. Has the statistical analysis been performed appropriately and rigorously? 

Reviewer #1: I Don't Know

Reviewer #2: Yes

Reviewer #3: No

Reviewer #4: Yes

3. Have the authors made all data underlying the findings in their manuscript fully available?

Reviewer #1: Yes

Reviewer #2: Yes

Reviewer #3: Yes

Reviewer #4: Yes

4. Is the manuscript presented in an intelligible fashion and written in standard English?

Reviewer #1: Yes

Reviewer #2: Yes

Reviewer #3: Yes

Reviewer #4: Yes

5. Review Comments to the Author

Reviewer #1: Thank you for sending!

In general, the present manuscript by Hiroto Y et al. described the transcriptome analysis after sevoflurane exposure in mice and specifically concluded some important regulators such as KLF4 in sevoflurane anesthesia. The aim of this study is of interest to relevant researchers and also important to clinical anesthesia. However, there are a lot of fundamental concerns both from experimental designs and over-interpretation of the data:

1. As a control, sleeping mice were used. The relational is acceptable. However, many factors can affect the comparability between groups, including but not limited to sleeping length vs. 3h-exposure of sevoflurane; did you consider about physiological circadian rhythms between these groups of mice? So, the data between sevoflurane anesthesia and baseline is more reliable than the difference between anesthesia and sleeping.

2. Only a small number of mice used in each group, so the signal to noise ratio may be not good to get a solid conclusion.

3. Single-cell sequencing is already common. The technique of RNA-seq is not state-of-art method for such comparisons. In introduction, there is no any description of already published studies about transcriptome after anesthesia, which is critical for relational of the study: what is already known and what is need to know?

4. The last and the most significant concern is that: the authors declared that KLF4 is a specific responsible transcription factor that potentially promotes angiogenesis and induces the appearance of undifferentiated neural cells. All these conclusions are completely based on data analysis. Without any actual measurement of angiogenesis and neural development after sevoflurane exposure, and did not design any intervention aiming these transcription factor cannot conclude such statements. Overall, the conclusion of the present manuscript is much over-interpreted, which need substantial revision. There is no solid causality between the transcription factor and sevoflurane exposure, as well as the proposed outcomes after anesthesia.

Minors:

1. The short title is not correct;

2. Why the mice exposed to sevoflurane with 3 hours? Is there any exposure-time dependent effect?

3. In results part, there are too much re-descriptions like methods.

Reviewer #2: The manuscript " Transcriptome analysis of sevoflurane exposure effects at the different brain regions" submitted by Hiroto Yamamoto et al. uses RNA-seq to analyze the differential gene expression from different brain regions. They conclude that Klf4 was upregulated by sevoflurane inhalation in whole brain. KLF4 might promote angiogenesis and cause the appearance of undifferentiated neural cells by transcriptional regulation. Overall, the results are interesting. However, the questions below need to be clarified.

1. Now that Klf4 is upregulated whole brain, the expression of Klf4 in protein levels need to be added through western blot or immunohistochemistry in four brain regions which will confirm your conclusion.

2. Anesthetics including sevoflurane can cause POCD, especially in elderly patients. Why did the authors use 8-week-old mice instead of aged mice?

3. In fact, only four brain regions were used to analyze the expression of different genes, however, the conclusion was the expression in whole brain. These four brain regions do not represent the whole brain. What about the brainstem and olfactory bulb?

4. About the treatment of control group mice, why did the control group mice use normal air instead of 40%O2?

5. Delete 138 lines of redundant “analysis”

Reviewer #3: Re: Review of PONE-D-20-21488

Thank you for the opportunity to review the manuscript “Transcriptome analysis of sevoflurane exposure effects at the different brain regions” by Yamamoto et al.

The author performed animal experiments and RNA-seq analysis to focus on a simultaneous analysis of the effects of sevoflurane on the gene expression changes in multiple anatomical sites of the brain.

This is indeed an interesting topic for review and discussion amongst the international community. I have listed my comments/concerns below.

(1) The sample size of this study (n=6; sevo group, n=3 vs ctl group, n=3) is too small to draw strong conclusions from the current data.

(2) The transcriptome array data of sleeping mice used in this study from existing database. Sleeping mice should be set as a group in your study, if possible.

(3) Problems on scientific writing: discussion of the results in the Results section would better fit into the discussion section.

For example: KLF4 is a famous transcription factor for sustaining the undifferentiated state of iPS cells, known as the “Yamanaka factor”. NES is a protein marker of neural stem cells and rarely expressed in differentiated neural cells. The upregulation of these genes suggest the possibility of induction of the appearance of undifferentiated neural cells by sevoflurane [21-24]. etc.) Pages 12, Lines 192-196; Pages 13, Lines 203-204; Pages 14, Lines 220-221; Pages 15, Lines 239-241; etc.

(4) Since gene expression in the hippocampus was the most-influenced in sevoflurane group based on your results, why not compare the transcriptome array data of the hippocampus of sleeping mice with sevoflurane exposure? Only a comparison was made with the transcriptome array data of the cerebral cortices of sleeping mice in this study.

(5) It is interesting but questionable that very short (3 hr) sevoflurane exposure upregulates KLF4. Furthermore, there may be still some doubt about whether KLF4 upregulated by sevoflurane exposure are really associated with the upregulation of angiogenesis and appearance of undifferentiated neural cells in whole brain. Also, the author did not evaluate protein expression changes for these genes, and only three of samples were used. Therefore, the evidence for the Conclusion is insufficient in the present results.

Reviewer #4: A well designed and interesting study investigating the effects of sevoflurane on gene expression in various regions in the brain. The results suggest that Klf4 dysregulation is responsible for promoting angiogenesis and for the appearance of undifferentiated neural cells.

While the authors state that the data is available through the DNA Data Bank Japan, I could not find the enrty. Perhaps it is private until publication? Please do ensure that this will be publicly available as this will be a valuable resource for the research community.

6. PLOS authors have the option to publish the peer review history of their article (what does this mean?). If published, this will include your full peer review and any attached files.

Reviewer #1: No

Reviewer #2: No

Reviewer #3: No

Reviewer #4: No

---

## [Author Response · Author response to Decision Letter 0]

21 Oct 2020

Reviewer #1’s comment #1

As a control, sleeping mice were used. The relational is acceptable. However, many factors can affect the comparability between groups, including but not limited to sleeping length vs. 3h-exposure of sevoflurane; did you consider about physiological circadian rhythms between these groups of mice? So, the data between sevoflurane anesthesia and baseline is more reliable than the difference between anesthesia and sleeping.

Our response to Reviewer #1’s comment #1

We agree with this comment. As the reviewer pointed out, this experiment would be expected to yield different results for a variety of factors, including circadian rhythms, and a comparison of our anesthesia data with the sleep data in another study might contain a fragility. If a common factor was found in these data and considered noteworthy, we could have pursued it more deeply, but we did not engage deeply in this comparison because our results showed that the expression patterns were totally different. We would like to adopt this comparison as the data showing that the changes focusing on in this study are specific to anesthesia, and we revised the text in the Discussion section accordingly. However, the overall argument does not change without this comparison. Therefore, if the reviewers appreciate that this paper is better off without this data, please reiterate that, as we will be withdrawing this comparison data. 

LINE 345-347

Discussion

[original]

The comparison of gene expressions in the brains of sleeping mice revealed that gene expression changes were specific to the brains exposed to sevoflurane.

[revised (red letters show the added descriptions)]

Detailed analysis between anesthesia and sleep is difficult because of the different experimental conditions, but at least in this comparison, gene expression changes in the brain exposed to sevoflurane showed a pattern that was very different from that of sleep.

Reviewer #1’s comment #2

Only a small number of mice used in each group, so the signal to noise ratio may be not good to get a solid conclusion.

Our response to Reviewer #1’s comment #2

We appreciate your comment. As rightly pointed out, the number of samples is a limitation of this study. However, we concluded that increasing replicates did not significantly change the results because the the PCA plot showed high reproducibility between triplicate (S1 Fig B). In this regard, we added the description of the limitation as follows:

LINE 361-364

Discussion

[original]

As the limitation of this study, only three of samples were used.

[revised (red letters show the added descriptions)]

As a limitation of this study, only three of samples were used. However, we concluded that increasing replicates did not significantly change the results because of the high reproducibility between triplicates, supported by the PCA plot (S1 Fig B)

Reviewer #1’s comment #3

Single-cell sequencing is already common. The technique of RNA-seq is not state-of-art method for such comparisons. In introduction, there is no any description of already published studies about transcriptome after anesthesia, which is critical for relational of the study: what is already known and what is need to know?

Our response to Reviewer #1’s comment #3

We appreciate your comment. As mentioned, single-cell sequencing is indeed a cutting edge and reliable method, and we have implemented it in our lab in various ways in other studies. On the other hand, it requires costly and time-consuming experiments. This was an introductory study, and in order to obtain a complete picture of the effects on the brain under anesthesia, we prioritized the analysis of many sites with bulk RNA-seq. We believe that the present objective could be achieved with bulk RNA-seq. However, as stated by the reviewer, more insights may be obtained with single-cell RNA-seq. In particular, analyses that combine single-cell RNA-seq and location information, such as Slide-seq (Rodriques et al., 2019. Science), may be useful to elaborate this study. These points have been added in the Discussion section as follows. 

LINE 352-358

Discussion

[original]

For precise understanding of the specific roles of KLF4 induced by sevoflurane, chromatin immunoprecipitation analysis of Klf4 and histone markers, such as H3K9me3 and H3K27Ac in each part of brain are needed. Nevertheless, our analysis results indicated the importance of KLF4 as a candidate regulator of the effects caused by sevoflurane inhalation.

[revised (red letters show the added descriptions)]

For a precise understanding of the specific role of KLF4 induced by sevoflurane, chromatin immunoprecipitation analysis of KLF4 and histone markers, such as H3K9me3 and H3K27ac in each part the of brain is needed. Furthermore, experimental methods that combine single-cell RNA-seq and location information such as Slide-seq may provide more useful information [33]. Nevertheless, our analysis results indicated the importance of KLF4 as a candidate regulator of the effects caused by sevoflurane inhalation.

Reference

[33] Rodriques SG, Stickels RR, Goeva A, Martin CA, Murray E, Vanderburg CR, et al. Slide-seq: A scalable technology for measuring genome-wide expression at high spatial resolution. Science. 2019;363(6434):1463-7.

As for our response to the second part of the comment. There have been several reports on transcriptome analysis of post-anesthesia brain samples, but there have been no reports of simultaneous RNA-seq on multiple sites to see the whole picture, as done in our report. In the introduction, we cited such studies and described the differences between those and ours as follows.

LINE 63-69

Introduction

[original]

Furthermore, although many reports have evaluated effects of sevoflurane, the analyses were focused only on limited parts of the brain; hence, comparison of the effects of sevoflurane between multiple anatomical sites at the same time is difficult.

[revised (red letters show the revised descriptions)]

Furthermore, although some reports have evaluated the effects of sevoflurane using transcriptome analysis, these studies focused only on limited parts of the brain. Hayase et al. reported that the increase in dopamine activity in the hippocampus due to inhalation of sevoflurane might be related to postoperative nausea, and Mori et al. reported circadian gene variations in the suprachiasmatic nucleus after sevoflurane inhalation [10,11]. However, we thought that by comparing many regions at once and extracting genes that might play common role in all regions, we could focus on genes that are important with regard to the whole brain.

Reference

[10] Hayase T, Tachibana S, Yamakage M. Effect of sevoflurane anesthesia on the comprehensive mRNA expression profile of the mouse hippocampus. Med Gas Res. 2016;6(2):70-6.

[11] Mori K, Iijima N, Higo S, Aikawa S, Matsuo I, Takumi K, et al. Epigenetic suppression of mouse Per2 expression in the suprachiasmatic nucleus by the inhalational anesthetic, sevoflurane. PLoS One. 2014;9(1):e87319.

Reviewer #1’s comment #4

The last and the most significant concern is that: the authors declared that KLF4 is a specific responsible transcription factor that potentially promotes angiogenesis and induces the appearance of undifferentiated neural cells. All these conclusions are completely based on data analysis. Without any actual measurement of angiogenesis and neural development after sevoflurane exposure, and did not design any intervention aiming these transcription factor cannot conclude such statements. Overall, the conclusion of the present manuscript is much over-interpreted, which need substantial revision. There is no solid causality between the transcription factor and sevoflurane exposure, as well as the proposed outcomes after anesthesia. 

Our response to Reviewer #1’s comment #4

We appreciate and totally agree with this comment. It was difficult to determine whether the upregulation of Klf4 by sevoflurane regulated angiogenesis or the appearance of undifferentiated neural cells, only from our RNA-seq analysis. We intended to claim that multiple analyses based on RNA-seq of the brain exposed to sevoflurane suggested some important roles of Klf4. It is worth noting that the possibility of upregulation of angiogenesis supported the report by Jiang et al. as mentioned in the discussion. We speculated that these results may be relevant to post-anesthetic events, such as POD, POCD, and emergence agitation, but further study is needed for its validation. Based on these considerations, we changed the discussion and conclusion as follows:

LINE 49-50

Abstract

[original]

Klf4 was upregulated by sevoflurane inhalation in whole brain. KLF4 might promote angiogenesis and cause the appearance of undifferentiated neural cells by transcriptional regulation. The roles of KLF4 might be key to elucidating the mechanisms of sevoflurane induced functional modification in the brain.

[revised]

Klf4 was upregulated by sevoflurane inhalation in the mouse brain. The roles of KLF4 might be key to elucidating the mechanisms of sevoflurane induced functional modification in the brain.

LINE 76-79

Introduction

[original]

Herein, we report our success in identifying KLF4 as a specific responsible transcription factor that potentially promotes angiogenesis and induces the appearance of undifferentiated neural cells.

[revised (red letters show the revised descriptions)]

Herein, we report sevoflurane-induced gene expression change patterns in the mouse brain and that KLF4 emerged as a specific transcription factor that potentially promoted angiogenesis and induced the appearance of undifferentiated neural cells.

Reviewer #1’s minor comment #1

The short title is not correct;

Our response to Reviewer #1’s minor comment #1

We appreciate your comments. In the original version, the book title and the short title were the same. We have given it the following short title.

[revised]

Short title: Effects of sevoflurane on brain

Reviewer #1’s minor comment #2

Why the mice exposed to sevoflurane with 3 hours? Is there any exposure-time dependent effect?

Our response to Reviewer #1’s minor comment #2

We appreciate your comment. Anesthetizing mice for 3h is a common method used in anesthesia experiments, and we followed this method. We also took into account the fact that general anesthesia for 3h is very common in clinical practice. We agree that it is interesting to compare the variable time duration of anesthesia, but we put more weightage on the comparison between multiple parts of the brain under anesthesia in this report.

Reviewer #1’s minor comment #3

In results part, there are too much re-descriptions like methods.

Our response to Reviewer #1’s minor comment #3

We appreciate this comment. There were certainly many re-descriptions in our original version. We deleted the sentences indicated by the reviewer from the result part and extended the discussion part as follows.

LINE 204-205

Results

[revised (red letters show the revised description)]

Sevoflurane inhalation upregulated transcription factors such as Klf4 in all sampled parts (Fig. 2B). The expression level of Klf4 was >2.5 times higher than that in the control mice. Furthermore…

LINE 210-212

Results

[revised]

The transcription factors KLF4 and KLF2, as well as EDN1, CCN1, and ADAMTS1, were annotated to the GO terms “angiogenesis” and “response to wounding” (S3 Table).

LINE 225-228

Results

[revised]

Moreover, the heatmap showed that sevoflurane inhalation downregulated the genes annotated as “head development” in all sampled parts of brain, and those annotated as “axon development” or “synapse organization” in several parts (Fig. 3D and S4 Table).

LINE 244-246

Results

[revised]

Moreover, by comparing genes upregulated and downregulated in all parts of the brain exposed to sevoflurane, we found that all the genes except Edn1 were completely expressed differently (Fig.4C and D).

<Reviewer #2’s comment #1>

Now that Klf4 is upregulated whole brain, the expression of Klf4 in protein levels need to be added through western blot or immunohistochemistry in four brain regions which will confirm your conclusion.

Our response to Reviewer #2’s comment #1

We appreciate the reviewer’s comment. First, we performed immunohistochemistry for the cerebral cortex and hippocampus of mice exposed to sevoflurane. As a result, we confirmed distinct upregulation of Klf4 in the nucleus of cells in the cerebral cortex. On the other hand, the basal expression of Klf4 was high in the hippocampus, and we could not observe the upregulation of Klf4 in hippocampus with immunohistochemistry. Therefore, we performed western blotting for the hippocampus, and it showed that the expression of Klf4 was upregulated to some extent. These data are submitted as S2 Fig A and B. From these results, we concluded that sevoflurane inhalation caused the upregulation of Klf4. We have revised sentences as follows:

LINE 274-283

Results

[revised (red letters show the added descriptions in the revised version)]

Finally, in the hippocampus, the target genes of 14 transcription factors were downregulated (Fig.5H). These results indicate that Klf4 plays some important roles in gene expression in brains exposed to sevoflurane. To validate the upregulation of Klf4, we performed immunohistochemical analysis for the cerebral cortex and hippocampus. As a result, we observed that the expression of Klf4 was strongly upregulated in the nucleus of cells in the cerebral cortex of mice exposed to sevoflurane. On the other hand, nucleus in neural cells of hippocampus in both control mice and mice exposed to sevoflurane showed high expression of Klf4, and no significant changes were observed in immunohistochemical analysis (S2 Fig A). Based on these results, we performed western blotting analysis to validate the upregulation of Klf4 in the hippocampus, showing a certain upregulation of Klf4 (S2 Fig B).

LINE 373-375

Discussion

[original]

For further understanding, proteomic analysis of brains with sevoflurane inhalation and pathological assessment of more samples and oxygen saturation assessment in mice are needed. Nevertheless, our strategies should be better choices for grabbing the whole image of brain activities under an anesthetized condition.

[revised (red letters show the revised description: the description about lack of pathology was removed)]

Oxygen saturation assessment in mice may provide more reliable results. Nevertheless, our strategy should include better choices for obtaining the whole image of brain activities under anesthetized condition.

Reviewer #2’s comment #2

Anesthetics including sevoflurane can cause POCD, especially in elderly patients. Why did the authors use 8-week-old mice instead of aged mice?

Our response to Reviewer #2’s comment #2

We appreciate your comment. When we focused on the pathology of peri-operative events such as POCD, the use of elderly mice would have been more suitable. However, temporary confusion such as emergence agitation is observed more frequently in young patients. Furthermore, the effects of volatile anesthesia on the developing brain is receiving attention these days. Based on these perspectives, we adopted eight weeks old mice, which is common for anesthetic experiments, and tried to obtain the whole image of the influences of sevoflurane inhalation. The emphasis of the relativity between sevoflurane inhalation and POCD or POD may have led to misunderstandings of our intention. Therefore, we have corrected as following:

LINE 57-60

Introduction

[original]

Sevoflurane is a most frequently used volatile anaesthetics in general anaesthesia. Some reports discussed the peri-operative adverse effects of sevoflurane such as post-operative delirium and cognitive disorders, although whether anesthetics themselves cause peri-operative adverse effects is still controversial [1-3].

[revised (red letters show the revised descriptions)]

Sevoflurane is the most frequently used volatile anesthetic in general anaesthesia. Some reports discussed the perioperative adverse effects of sevoflurane, such as emergence agitation, postoperative delirium, and cognitive disorders, although whether anesthetics themselves cause perioperative adverse effects is still controversial [1-3].

Reviewer #2’s comment #3

In fact, only four brain regions were used to analyze the expression of different genes, however, the conclusion was the expression in whole brain. These four brain regions do not represent the whole brain. What about the brainstem and olfactory bulb?

Our response to Reviewer #2’s comment #3

We appreciate with the reviewer’s comment. We also think that the expression “whole brain” is over-interpretation. Therefore, we have changed the phrases of “whole brain” to “all sampled regions”

Reviewer #2’s comment #4

About the treatment of control group mice, why did the control group mice use normal air instead of 40%O2? 

Our response to Reviewer #2’s comment #4

We appreciated with the reviewer’s comment. The O2 concentration in the anesthetized group in our experiment was 40%, which seemed to be a common condition for both animal experiments and clinical practices. We also took into account that Ohe et al. performed in the same condition (2.5%sev/40%O2), and murine O2 saturation was stable in 95-100% (Ohe et al, Neuroscience Letters 2011). Exposure to the same concentration of O2 for control or anesthetized mice may result in lower O2 blood concentration in anesthetized mice, since the respiration of anesthetized mice will be suppressed. Based on this concern, controlling O2 concentration while monitoring murine O2 saturation might be a better resolution. Therefore, we changed the description of the limitation for O2 saturation as follows:

LINE 367-373

Discussion

[original]

None of the genes related to hypoxic reaction, including Hif1a and Arnt, were detected in our analyses in gene expression changes, supporting the exclusion of the possibility of hypoxia in our experimental condition (S1 Table).

[revised (red letters show the added descriptions)]

None of the genes related to hypoxic reaction, including Hif1a and Arnt, were detected in our analyses of gene expression changes, supporting the exclusion of the possibility of hypoxia in our experimental conditions (S1 Table). Conversely, oxygen saturation might have been higher in the anesthetized group than in the control group, which was allowed to spend time in room air, and since we did not measure oxygen saturation, it is possible that subtle differences in oxygen saturation existed and that this might have affected the results. Oxygen saturation assessment in mice may provide more reliable results.

Reviewer #2’s comment #5

5. Delete 138 lines of redundant “analysis”

Our response to Reviewer #2’s comment #5

As the reviewer pointed out, it was our mistake. We revised it.

Reviewer #3’s comment #1

The sample size of this study (n=6; sevo group, n=3 vs ctl group, n=3) is too small to draw strong conclusions from the current data.

Our response to Reviewer #3’s comment #1

We appreciate this comment. Another reviewer also pointed out the same thing. As the reviewers pointed out, the number of samples is a limitation of this study. However, we concluded that increasing replicates did not significantly change the results because of the reproducibility of the PCA plot (Fig. S1B) between the samples. In this regard, we have added the description of the limitation as follows:

LINE 361-364

Discussion

[original]

As the limitation of this study, only three of samples were used.

[revised (red letters show the added descriptions)]

As a limitation of this study, only three of samples were used. However, we concluded that increasing replicates did not significantly change the results because of the high reproducibility between triplicates, supported by the PCA plot (S1 Fig B).

Reviewer #3’s comment #2

The transcriptome array data of sleeping mice used in this study from existing database. Sleeping mice should be set as a group in your study, if possible.

Our response to Reviewer #3’s comment #2

Another reviewer also has noted the same thing, and we agree with this comment. This experiment was expected to yield different results for a variety of factors, and a comparison of our anesthesia data with the sleep data in another study might contain a fragility. It would have been best if we could do the experiment ourselves under the right conditions, but that sleep experiment seemed to require certain specialized skills and we concluded it would be difficult to obtain accurate data on our own. If a common factor was found in these data and considered noteworthy, we could have pursued it in more depth, but we did not get too deep into this comparison, because our results showed that the genes operated in a rather different pattern altogether. We would like to adopt this comparison as the data showing that the changes we focused on in this study were specific to anesthesia, and we revised the text in the discussion accordingly. However, the overall argument does not change without this comparison. Therefore, if the reviewers appreciate that this paper is better off without this data, please reiterate that, as we will be withdrawing this comparison data. 

LINE 345-347

Discussion

[original]

The comparison of gene expressions in the brains of sleeping mice revealed that gene expression changes were specific to the brains exposed to sevoflurane.

[revised]

Detailed analysis between anesthesia and sleep is difficult because of the different experimental conditions, but at least in this comparison, gene expression changes in the brain exposed to sevoflurane showed a pattern that was very different from that of sleep.

Reviewer #3’s comment #3

Problems on scientific writing: discussion of the results in the Results section would better fit into the discussion section. For example: KLF4 is a famous transcription factor for sustaining the undifferentiated state of iPS cells, known as the “Yamanaka factor”. NES is a protein marker of neural stem cells and rarely expressed in differentiated neural cells. The upregulation of these genes suggest the possibility of induction of the appearance of undifferentiated neural cells by sevoflurane [21-24]. etc.) Pages 12, Lines 192-196; Pages 13, Lines 203-204; Pages 14, Lines 220-221; Pages 15, Lines 239-241; etc.

Our response to Reviewer #3’s comment #3

We appreciate the reviewer’s comment. Another reviewer also had noted the same thing, with the comment that there is a lot of re-description. There was certainly a lot of re-descriptions both in both the Results and Discussion sections. We have revised some descriptions in the results section that have the same meaning in the discussion section as follows (the following lists of revisions are the same lists in response to another reviewer):

LINE 204-205

Results

[revised (red letters show the revised description)]

Sevoflurane inhalation upregulated transcription factors such as Klf4 in all sampled parts (Fig. 2B). The expression level of Klf4 was >2.5 times higher than that in the control mice. Furthermore…

LINE 210-212

Results

[revised]

The transcription factors KLF4 and KLF2, as well as EDN1, CCN1, and ADAMTS1, were annotated to the GO terms “angiogenesis” and “response to wounding” (S3 Table).

LINE 225-228

Results

[revised]

Moreover, the heatmap showed that sevoflurane inhalation downregulated the genes annotated as “head development” in all sampled parts of brain, and those annotated as “axon development” or “synapse organization” in several parts (Fig. 3D and S4 Table).

LINE 244-246

Results

[revised]

Moreover, by comparing genes upregulated and downregulated in all parts of the brain exposed to sevoflurane, we found that all the genes except Edn1 were completely expressed differently (Fig.4C and D).

Reviewer #3’s comment #4

Since gene expression in the hippocampus was the most-influenced in sevoflurane group based on your results, why not compare the transcriptome array data of the hippocampus of sleeping mice with sevoflurane exposure? Only a comparison was made with the transcriptome array data of the cerebral cortices of sleeping mice in this study.

Our response for Reviewer #3’s comment #4

We appreciate your comment. We also think it would be interesting to compare with the transcriptome array data of the hippocampus of sleeping mice. However, we could not find the depository data of hippocampus for comparison under the same conditions. The sleep experiments need expertized skills, and it was difficult to perform the experiment at the same quality by ourselves. 

Reviewer #3’s comment #5

It is interesting but questionable that very short (3 hr) sevoflurane exposure upregulates KLF4. Furthermore, there may be still some doubt about whether KLF4 upregulated by sevoflurane exposure are really associated with the upregulation of angiogenesis and appearance of undifferentiated neural cells in whole brain. Also, the author did not evaluate protein expression changes for these genes, and only three of samples were used. Therefore, the evidence for the Conclusion is insufficient in the present results.

Our response for Reviewer #3’s comment #5

We appreciate the reviewer’s comment. We were also concerned whether the exposure to sevoflurane for three h was too short to cause changes in protein expression. Another reviewer was also concerned about protein level variation. Therefore, we performed immunohistochemistry of the cerebral cortex and hippocampus of mice exposed to sevoflurane. As a result, we confirmed distinct upregulation of Klf4 in the nucleus of cells in the cerebral cortex. On the other hand, the basal expression of Klf4 was high in the hippocampus, and we could not observe the upregulation of Klf4 in the hippocampus with immunohistochemistry. Therefore, we performed western blotting for the hippocampus, and it showed that the expression of Klf4 was upregulated to some extent. These data are submitted as S2 Fig A and B. From these results, we concluded that sevoflurane inhalation caused the upregulation of Klf4. We did not evaluate changes in protein expression of its downstream genes, but Klf4 usually functioned as a transcription regulator, and changes in candidate downstream genes at the RNA level might be the supporting data of transcription regulatory roles of Klf4 with sevoflurane exposure. Based on these considerations, we changed the descriptions as follows: 

LINE 274-283

Results

[revised (red letters show the added descriptions in revised version)]

Finally, in the hippocampus, the target genes of 14 transcription factors were downregulated (Fig.5H). These results indicate that Klf4 plays some important roles in gene expression in brains exposed to sevoflurane. To validate the upregulation of Klf4, we performed immunohistochemical analysis for the cerebral cortex and hippocampus. As a result, we observed that the expression of Klf4 was strongly upregulated in the nucleus of cells in the cerebral cortex of mice exposed to sevoflurane. On the other hand, nucleus in neural cells of hippocampus in both control mice and mice exposed to sevoflurane showed high expression of Klf4, and no significant changes were observed in immunohistochemical analysis (S2 Fig A). Based on these results, we performed western blotting analysis to validate the upregulation of Klf4 in the hippocampus, showing a certain up-regulation of Klf4 (S2 Fig B).

Reviewer #4’s comment

While the authors state that the data is available through the DNA Data Bank Japan, I could not find the enrty. Perhaps it is private until publication? Please do ensure that this will be publicly available as this will be a valuable resource for the research community.

Our response for Reviewer #4’s comment

We appreciate your comment. We intend to disclose the RNA-Seq data and publish it in the DNA Data Bank Japan after acceptance. Please let us know if the deposit data should be published before the acceptance of this paper.

---

## [Decision Letter · Decision Letter 1]

13 Nov 2020

Transcriptome analysis of sevoflurane exposure effects at the different brain regions

PONE-D-20-21488R1

Dear Dr. Asahara,

We’re pleased to inform you that your manuscript has been judged scientifically suitable for publication and will be formally accepted for publication once it meets all outstanding technical requirements.

Kind regards,

Wataru Nishimura, M.D., Ph.D.

Academic Editor, PLOS ONE / Dept. of molecular biology, IUHW school of medicine 

Additional Editor Comments (optional):

Reviewers' comments:

Reviewer's Responses to Questions

**Comments to the Author**

1. If the authors have adequately addressed your comments raised in a previous round of review and you feel that this manuscript is now acceptable for publication, you may indicate that here to bypass the “Comments to the Author” section, enter your conflict of interest statement in the “Confidential to Editor” section, and submit your "Accept" recommendation.

Reviewer #1: (No Response)

Reviewer #2: All comments have been addressed

2. Is the manuscript technically sound, and do the data support the conclusions?

Reviewer #1: Yes

Reviewer #2: Yes

3. Has the statistical analysis been performed appropriately and rigorously? 

Reviewer #1: Yes

Reviewer #2: Yes

4. Have the authors made all data underlying the findings in their manuscript fully available?

Reviewer #1: Yes

Reviewer #2: No

5. Is the manuscript presented in an intelligible fashion and written in standard English?

Reviewer #1: Yes

Reviewer #2: Yes

6. Review Comments to the Author

Reviewer #1: Thank you for the revision and additional experiments. Generally, the revisions have almost addressed my comments. I only have one suggestion, as mentioned by authors themselves, the comparison between sevoflurane anesthesia and sleep do not make a big sense; so, please consider to remove such data or move to supplementary data.

Reviewer #2: The authors have answered all my concerns about the manuscript. I agree to accept the paper for publication.

7. PLOS authors have the option to publish the peer review history of their article (what does this mean?). If published, this will include your full peer review and any attached files.

Reviewer #1: **Yes: **Cheng Zhou

Reviewer #2: No

---

## [Editor Report · Acceptance letter]

23 Nov 2020

PONE-D-20-21488R1 

Transcriptome analysis of sevoflurane exposure effects at the different brain regions 

Dear Dr. Asahara:

I'm pleased to inform you that your manuscript has been deemed suitable for publication in PLOS ONE. Congratulations! Your manuscript is now with our production department. 

Kind regards, 

on behalf of

Dr. Wataru Nishimura 

Academic Editor

PLOS ONE